# The Effect of Childhood Obesity or Sarcopenic Obesity on Metabolic Syndrome Risk in Adolescence: The Ewha Birth and Growth Study

**DOI:** 10.3390/metabo13010133

**Published:** 2023-01-16

**Authors:** Hyunjin Park, Seunghee Jun, Hye-Ah Lee, Hae Soon Kim, Young Sun Hong, Hyesook Park

**Affiliations:** 1Department of Preventive Medicine, College of Medicine, Ewha Womans University, Seoul 07804, Republic of Korea; 2Graduate Program in System Health Science and Engineering, Ewha Womans University, Seoul 03760, Republic of Korea; 3Clinical Trial Center, Ewha Womans University Mokdong Hospital, Seoul 07985, Republic of Korea; 4Department of Pediatrics, College of Medicine, Ewha Womans University, Seoul 07804, Republic of Korea; 5Department of Internal Medicine, College of Medicine, Ewha Womans University, Seoul 07804, Republic of Korea

**Keywords:** adolescent, metabolic syndrome, childhood obesity, PsiMS, cMetS, SPISE

## Abstract

The prevalence of obesity and metabolic syndrome (MetS) in the pediatric population has increased globally. We evaluated the impact of childhood obesity and sarcopenic obesity on the risk of MetS in adolescence using the Ewha Birth and Growth Cohort study data. In this study, we analyzed data from 227 participants who were followed up at the ages of 7–9 and 13–15 years. Overweight and obesity were defined as a body mass index of the 85th percentile or higher based on national growth charts, and sarcopenic obesity was defined using body composition data. Metabolic diseases in adolescence were identified by calculating the pediatric simple metabolic syndrome score (PsiMS), continuous metabolic syndrome score (cMetS), and single-point insulin sensitivity estimator (SPISE) as MetS indices. The prevalence of overweight was approximately 15% at both 7–9 and 13–15 years old, and that of sarcopenic obesity (7–9 years old) was 19.5%. Boys aged 13–15 years had a significantly larger waist circumference (WC) and higher systolic blood pressure (SBP) than girls. The MetS indices (PsiMS, cMetS, and SPISE) showed no significant differences by gender. Overweight and sarcopenic obese people have a higher overall risk of MetS components than normal people. The overweight group had a significantly higher prevalence of PsiMS and cMetS than the normal group, while the SPISE was significantly lower and the MetS indicator was worse in the overweight group than in the normal group. Similar results were obtained in the group with sarcopenic obesity. Both overweight and sarcopenic obesity remained significantly associated with MetS indicators, even after adjusting for covariates. Furthermore, metabolic health assessed by the cMetS in adolescence was affected not only by childhood overweight but also by adolescence, which showed an interaction effect. The results of this study emphasize the importance and need for early detection of childhood obesity and effective public health interventions.

## 1. Introduction

The high prevalence of obesity in children and adolescents is a global public health problem [1]. Worldwide, the prevalence of overweight among children and adolescents increased from 0.7% to 5.6% in girls and from 0.9% to 7.8% in boys between 1975 and 2016 [2]. In Korea, based on the Korea National Health and Nutrition Examination Survey (KNHANES), the prevalence of obesity in children and adolescents aged 6–18 years increased from 10.2% in 2010–2012 to 14.8% in 2019–2020 [3]. A domestic study estimated that about KRW 1363.8 billion in socioeconomic costs occur when adolescent obesity continues as adult obesity [4].

Obesity can be traced from childhood through adulthood [5] and contributes to adverse outcomes, including premature mortality and cardiometabolic disorders [6,7]. In addition, being overweight or obese is a key risk factor for the development of metabolic syndrome (MetS), and these characteristics promote the initiation and progression of chronic diseases, such as cardiovascular disease (CVD) and type 2 diabetes (T2DM) [1,8,9]. Metabolic syndrome is not only a problem of adulthood; it is already present in some children and adolescents.

Another problem with the recent increase in obesity is sarcopenic obesity, which refers to an unbalanced relationship between high fat mass and low muscle mass [10,11] and is usually described more often in adulthood. Children who have a normal body mass index (BMI) and appear non-obese may nevertheless have little muscle mass and more body fat compared to their peers [12]. Sarcopenia has been identified as a risk factor for insulin resistance and high metabolic risk in children and adolescents [13,14]. The co-occurrence of sarcopenia and obesity amplifies the risk of adverse health outcomes [15,16]. However, despite the importance of early detection of obesity in children and adolescents, few studies have attempted to do this.

Since childhood and adolescence are developmental stages, it is difficult to set one criterion for MetS, and continuous assessments are required to identify these risks. Identifying MetS in obese or overweight children accurately is the basis for strategies to prevent metabolic diseases and complications later in life. Several tools are used to define MetS in children and adolescents [17,18,19]. In this study, we evaluated the prospective associations of childhood and sarcopenic obesity with MetS risk in adolescence using the PsiMS, cMetS, and SPISE as MetS-related indices.

## 2. Methods

### 2.1. Study Participants

The study participants were sampled from the Ewha Birth and Growth Cohort Study, an ongoing birth cohort established at Mokdong Hospital, Ewha Womans University, Seoul, Republic of Korea, between 2001 and 2006. The details of this cohort have been reported elsewhere [20]. Regular check-ups were conducted at 3, 5, and 7 years of age and annually thereafter. Of the 940 children enrolled in the birth cohort, data were available on 582 children aged 7–9 years (292 boys and 290 girls) and 248 adolescents aged 13–15 years (121 boys and 127 girls). Of these children and adolescents, 233 participated in follow-up at the ages of 7–9 and 13–15 years. After excluding participants who did not have metabolic indicator data for the anthropometric and MetS components, the study ultimately analyzed 227 participants.

The Institutional Review Board (IRB) at Ewha Womans University Seoul Hospital approved the study protocol after receiving written informed consent from all participants and their parents or legal guardians (IRB number: SEUMC 2020-07-016-002).

### 2.2. Variables

All anthropometric and blood pressure (BP) measurements were done by trained researchers or nurses. Participants wearing light clothing and no shoes were measured to the nearest 0.1 cm using an automatic height scale (GL-150, G-Tech International Co., Ltd., Uijeongbu, Republic of Korea). Body mass index (BMI; kg/m^2^) was calculated as weight divided by height squared. Waist circumference (WC) was measured in the standing position during expiration at the mid-point between the inferior margin of the 12th rib and the upper margin of the iliac crest. BP was measured twice using an automatic blood pressure monitor (BPBIO320, InBody, Seoul, Republic of Korea) when the subject was in a stable state. The average of the two measurements was used. The mean arterial pressure (MAP) was calculated as: [(systolic BP − diastolic BP)/3] + diastolic BP.

At the time of follow-up, blood tests were performed after fasting for 9 h to obtain MetS index values such as triglyceride (TG), high-density lipoprotein cholesterol (HDL-C), and fasting blood sugar (FBG).

Body composition at 7–9 years old was measured via bioelectrical impedance (InBody 720, Biospace Co., Ltd., Seoul, Republic of Korea). From the InBody device, skeletal muscle mass (SMM) and body fat mass (BFM) were obtained. To calculate the muscle/fat ratio (MFR), SMM was divided by BFM (kg/kg).

As covariates, gender, monthly household income (<3 million KRW/3–5 million KRW/≥ 5 million KRW), mother’s education level (low/high), parental history of hypertension (yes/no), overeating (none/1–2 times per week/≥3 times per week), and moderate physical activity (yes/no) related to individual lifestyle were considered. If at least one of the parents reported a diagnosis of hypertension by a physician, it was considered a parental history of hypertension. The frequency of overeating was categorized as almost none, 1–2 times a week, and 3 or more times a week based on the data distribution.

### 2.3. Definition of Overweight and Sarcopenic Obesity

Participants were categorized as normal (<85th percentile) or overweight (≥85th percentile) according to their current BMI using gender- and age-specific criteria from the 2017 Korean National Growth Charts for children and adolescents [21]. Childhood (7–9 years) sarcopenic obesity was defined as 1.25 for boys and 1.1 for girls according to the MFR cutoff for children described by McCarthy et al. [22].

### 2.4. Definition of Metabolic Syndrome Index

In this study, MetS indicators were calculated between the ages of 13 and 15 to evaluate the metabolic risk in adolescence.

The PsiMS is a modified continuous MetS score for use in obese adolescents based on the original SIMS score [20]. It was calculated using the following formula: (2 × (waist/height (cm))) + (FBG (mg/dL)/100) + (TG (mg/dL)/150) + (Systolic BP (mmHg)/130) − (HDL-C (mg/dL)/40) [17].

Standardized cMetS values were calculated for the MetS components (BMI, FBG, TG, MAP, and HDL-C) using the Z-score method. Because HDL-C has an inverse association, the Z-scores of HDL-C were multiplied by −1. The cMetS was calculated using the Z-score of each component by applying the formula [BMI + FBG + TG + MAP – HDL-C], where a higher value predicts a relatively higher risk of MetS [18].

SPISE was calculated based on HDL-C and TG levels and BMI using the following equation: SPISE = 600 × HDL-C^0.185^/(TG^0.2^ × BMI^1.338^) [19]. This study did not evaluate the Insulin Sensitivity Index (ISI), Homeostasis Model Assessment for Insulin Resistance (HOMA-IR), or secretion (HOMA-β%) indicators that estimate insulin sensitivity and resistance.

### 2.5. Statistical Analysis

All variables were checked for normality and are presented as the mean ± standard deviation for continuous variables and the number of participants (%) for categorical variables. Because the TG level was not normally distributed, it was log-transformed for the analysis.

The mean differences in the MetS indices and MetS components according to basic characteristics were evaluated using the *t*-test and a generalized linear model. A univariate linear regression was used to evaluate the effect of overweight or sarcopenic obesity on the MetS index. Then, with adjustments for confounding variables, a multiple linear regression analysis was performed. The results are expressed as adjusted means with a 95% confidence interval (95% CI) and *p*-value. The interaction effect of overweight at the two observation points (at 7–9 and 13–15 years old) on the MetS index was evaluated using a generalized linear model.

Statistical significance was assessed at *p* < 0.05 with a two-tailed test. All statistical analyses were performed using SAS version 9.4 (SAS Institute, Cary, NC, USA).

## 3. Results

Table 1 shows the anthropometric and body composition characteristics of the participants who were followed up between the ages of 7–9 and 13–15 years. The 582 participants aged 7–9 years were balanced in terms of gender (boys, *n*= 292, and girls, *n* = 290). In the 13–15 years age group (*n* = 248), the proportions of boys (48.8%; *n* = 121) and 51.2%; *n* = 127) and girls were similar. The average BMI at age 7–9 years was 16.50 kg/m^2^, and 15.0% were overweight. The average BMI at age 13–15 years was 20.69 kg/m^2^, and 15.4% were overweight, similar to the 7-9-year-old group. Most of those who were overweight at age 7–9 years were also overweight at age 13–15 years. Among the subjects seen at 7–9 years of age, sarcopenic obesity was confirmed in 19.5% (78/582). Furthermore, at 7–9 years of age, sarcopenic obesity accounted for 84.4% of the overweight children.

The metabolic characteristics and health behaviors of the 227 participants aged 13–15 years are listed in Table 2. Boys had significantly higher WC (4.46 cm, *p* < 0.001) and systolic BP (3.52 mmHg, *p* < 0.05) than girls. The difference in TG showed marginal significance (4.18 in boys and 4.30 in girls, *p* = 0.05); the other MetS components (BMI, FBG, and HDL-C) did not differ by gender. Also, there was no significant difference between boys and girls in the MetS indicators (PsiMS, cMetS, and SPISE). Among health behavior factors, there was a significant difference between boys and girls in moderate physical activity (*p* < 0.05), but no difference was observed in other factors.

Figure 1 shows the mean differences between the MetS components in the group with normal-overweight and the group with normal sarcopenic obesity. Overall, the MetS components pose a higher risk for overweight and sarcopenic obesity children than in normal children.

Regarding the means of the MetS indices in each group (Table 3), the overweight group had significantly higher PsiMS and cMetS than the normal subjects (*p* < 0.001); the values were approximately 0.37 and 2.82 higher, respectively. The SPISE level was also significantly lower, and the MetS index was worse in the overweight (7.1) than in the normal group (10.1). Similarly, in sarcopenic obesity, PsiMS and cMetS were significantly higher than in the normal weight group (*p* < 0.01), while the SPISE level was significantly lower.

Table 4 shows the results of multiple linear regression analysis of the associations with MetS indices by overweight or sarcopenic obesity. Both overweight and sarcopenic obesity were significantly associated with MetS indices, even after adjusting for health behavior, parents’ socioeconomic level, and parental history of hypertension. Despite adjusting for various variables, the overweight group had a significantly increased cMetS, by 2.63, and a significantly decreased SPISE, by –2.64, compared to the normal group (both *p* < 0.001). In sarcopenic obesity, cMetS increased significantly (by 1.96) compared with the normal group in the adjusted model, and SPISE also showed a significant association (*p* < 0.001).

The interaction between overweight status at 7–9 and 13–15 years (Figure 2) was statistically significant (*p* for the interaction = 0.04 for cMetS). PsiMS and SPISE showed a trend toward an interaction effect (*p* < 0.1).

## 4. Discussion

Our findings suggest that childhood overweight and sarcopenic obesity are associated with MetS in early adolescence, regardless of the three indices reflecting MetS. Furthermore, the associations persisted after adjusting for various covariates. Among MetS indices, the effect of overweight status at the age of 7–9 years on cMetS in adolescence differed according to the overweight status at the age of 13–15 years, showing an additive interaction (*p* = 0.04). From a life-course perspective, obesity during childhood may adversely affect metabolic health later in adolescence. Therefore, our results support the need for appropriate interventions for obesity in childhood to reduce metabolic diseases later in life.

In children and adolescents, due to the low prevalence and loss of information resulting from binary assessments of MetS, several continuous indices have been proposed to reflect MetS risk [17,18,19]. The PsiMS is an accurate and efficient scoring system to assess and monitor the risk of adolescent MetS in research and clinical settings [17]. In a study of adolescents using the KNHANES data, the mean PsiMS was significantly higher in participants with MetS compared with those without MetS (3.24 vs. 1.93 in boys, 3.11 vs. 1.80 in girls; all *p* < 0.001). Further, PsiMS showed good performance in identifying the risk of MetS in adolescents (area under the curve = 0.958) [23]. In addition, cMetS can be used to assess the risk of MetS in children and adolescents, which is useful for pediatric research. In a previous study of participants aged 7–18 years, the mean cMetS increased with the number of MetS components and was sufficiently accurate and sensitive to predict MetS (area under the curve = 0.94) [18]. The validity of cMetS was confirmed in a study of 7–9-year-old students conducted in the United States [24]. SPISE has also been used as a surrogate marker of insulin sensitivity to predict MetS and screen for T2DM in adults [25]. In a study in which the participants had an average age of 16.8 years, SPISE was a better diagnostic tool for predicting MetS than other indicators (HOMA-IR or TG-HDL) for both male and female adolescents [26]. Additionally, a previous cohort study showed a positive correlation between SPISE and insulin sensitivity derived from a euglycemic insulin clamp test in adolescents [19].

Although some studies have evaluated the association between each MetS index and obesity [27], no study has assessed the effect of childhood obesity on the risk of MetS in early adolescence using various MetS-related indices. Also, no studies have evaluated childhood sarcopenic obesity or MetS in adolescence. Similar to this study, a previous study [27] found differences in MetS scores among normal, overweight, and obese children according to BMI categories aged 7–9 years. A cross-sectional study conducted in Beijing reported a strong association between obesity and MetS, defined based on the International Diabetes Federation (IDF) definition, in children aged 10–18 years [28]. In addition, when 200 children with overweight or obesity were followed for 6.5 years, a low SPISE index showed an OR value of 3.89 (95% CI = 1.65–9.13), regardless of major metabolic confounders such as gender and age [29]. In children, a low SPISE is significantly associated with metabolic abnormalities. Metabolic abnormalities in children and adolescents can predict MetS in adults [30,31,32]. In four large cohort studies (average follow-up, 22.3 years) participating in the International Childhood Cardiovascular Cohort (i3C) Consortium, cMetS in childhood was associated with a 2.14-fold (95% CI 1.19–3.85) higher risk of MetS in adults [33].

Children and adolescents with overweight and obesity are more likely to have cardiovascular risk factors such as high blood pressure, lipid metabolism disorders, and glucose metabolism disorders. This may lead to the development of non-communicable diseases in adulthood [33]. Therefore, as the burden of metabolic diseases increases in childhood, adolescence, and early adulthood, it is essential to predict them to prevent and monitor them. Despite its importance from a life-course perspective, many studies have examined childhood obesity and CVD risk in adulthood [33,34], but few have evaluated the relationship with disease susceptibility in adolescence. Therefore, further evaluations are required through prospective studies.

In line with previous studies, we observed a significant association between sarcopenic obesity and metabolic risk. On evaluating Korean children and adolescents (10–18 years old), the odds ratio (OR) of the risk of MetS increased significantly in sarcopenic obesity (OR = 8.28; 95% CI = 5.6–11.5) [35]. Another study of 16–17-year-old subjects had similar results (OR = 21.2; 95% CI 4.2–107.5 for boys; OR = 3.6; 95% CI 1.1–11.9 for girls) [36]. In this study, those with sarcopenic obesity had higher PsiMS and cMetS scores and a lower SPISE than participants of normal weight. In adults, sarcopenic obesity has a stronger association with MetS risk than either obesity or sarcopenia alone [37,38]. Sarcopenic obesity may carry a cumulative metabolic risk of both sarcopenia and obesity and may lead to worse metabolic outcomes than obesity alone. In this study, since most of the participants in the sarcopenic obese and overweight groups overlapped, there were limitations in terms of our ability to evaluate them in detail. Therefore, large-scale studies are needed to understand the effect of sarcopenic obesity on metabolic health. In addition, the ISI, HOMA-IR, and HOMA-β% indicators were not evaluated in this study and warrant further investigation.

Despite these limitations, our findings provide evidence of early health determinants associated with MetS-related risk, particularly those associated with the long-term effects of childhood obesity. Our data also highlight the public-health implications of childhood obesity in terms of adolescent health outcomes.

## 5. Conclusions

The birth cohort data provide evidence of a meaningful and significant association between childhood obesity and MetS risk in adolescence. The results emphasize the importance of early detection of childhood obesity and effective public health interventions.

## Figures and Tables

**Figure 1 metabolites-13-00133-f001:**
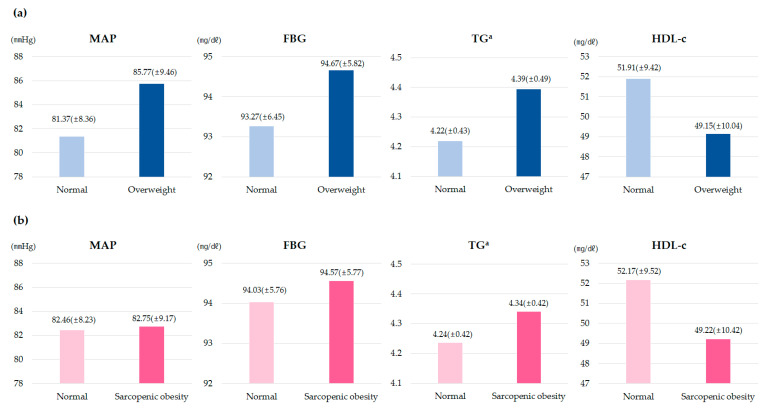
Distribution of metabolic risk factors among study subjects according to childhood obesity. MAP, mean arterial pressure; FBG, fasting blood glucose; TG, triglyceride; HDL-C, high-density lipoprotein cholesterol; (**a**) Subjects were classified as normal or overweight with the 85th percentile as the cut-off point; (**b**) sarcopenic obesity was defined as 1.25 for boys and 1.1 for girls. ^a^ Log transformation was performed because it was not normally distributed.

**Figure 2 metabolites-13-00133-f002:**
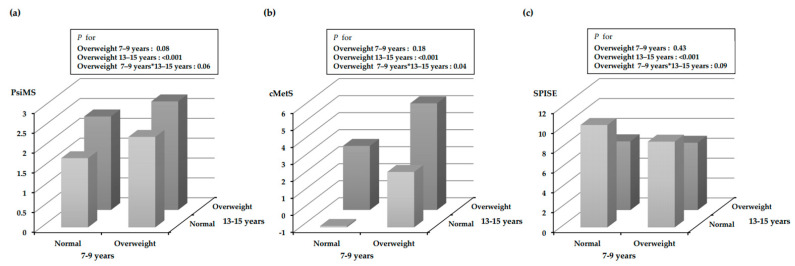
The interaction effect of overweight groups at two time points (at ages 7–9 and 13–15 years) on the metabolic syndrome index in adolescence. Adjusted for sex, household income, mother’s education level, parental history of hypertension, overeating, and moderate physical activity. BMI, body mass index; PsiMS, pediatric simple metabolic syndrome score; cMetS, continuous metabolic syndrome score; SPISE, single-point insulin sensitivity estimator. (**a**) The interaction between the 7–9 years BMI group and the 13–15 years BMI group on PsiMS; (**b**) The interaction between the 7–9 years BMI group and the 13–15 years BMI group on cMetS; (**c**) The interaction between the 7–9 years BMI group and the 13–15 years BMI group on SPISE.

**Table 1 metabolites-13-00133-t001:** Anthropometric and body composition characteristics of the study at 7–9 and 13–15 years follow-up.

Variables	7–9 Years (*n* = 582)	13–15 Years (*n* = 248)
Height (cm)	126.69 (±6.83)	160.41 (±6.59)
Weight (kg)	26.77 (±6.04)	53.42 (±10.62)
WC (cm)	56.28 (±7.01)	70.46 (±8.95)
BMI (kg/m^2^)	16.5 (±2.44)	20.69 (±3.34)
Overweight ^a^, n (%)	87 (15.0)	38 (15.4)
SMM (kg) ^b^	10.04 (±1.90)	NA
BFM (kg) ^b^	6.48 (±3.59)	NA
MFR (kg/kg) ^b^	1.90 (±0.90)	NA
Sarcopenic obesity ^c^, n (%)	78 (19.5)	NA

WC, waist circumference; BMI, body mass index; SMM, skeletal muscle mass; BFM, body fat mass; MFR, muscle fat ratio; NA, not applicable. Numerical values represent the mean (± standard deviation), while categorical values are represented by n (%). ^a^ Subjects were classified as normal or overweight, with the 85th percentile as the cut-off point suggested by national growth charts [21]. ^b^ Body composition was not measured at the age of 13–15 years. ^c^ Sarcopenic obesity was defined as 1.25 for boys and 1.1 for girls according to the criteria proposed by McCarthy et al. [22].

**Table 2 metabolites-13-00133-t002:** Characteristics of study subjects (at the age of 13–15 years).

Variable	Total (*n* = 227)	Boys (*n* = 111)	Girls (*n* = 116)	*p*-Value
WC (cm)	70.42 (±9.02)	72.7 (±10.35)	68.24 (±6.89)	<0.001
BMI (kg/m^2^)	20.71 (±3.41)	20.92 (±3.83)	20.51 (±2.95)	0.37
MAP (mmHg)	81.89 (±8.59)	82.92 (±8.95)	80.91 (±8.15)	0.08
SBP (mmHg)	110.44 (±11.55)	112.24 (±11.83)	108.72 (±11.04)	0.02
DBP (mmHg)	67.62 (±8.36)	68.26 (±8.90)	67.00 (±7.80)	0.26
FBG (mg/dL)	93.44 (±6.38)	93.65 (±5.72)	93.23 (±6.97)	0.62
log TG ^a^	4.24 (±0.44)	4.18 (±0.47)	4.30 (±0.41)	0.05
HDL-C (mg/dL)	51.58 (±9.51)	51.55 (±8.97)	51.6 (±10.05)	0.97
MetS index				
PsiMS	1.89 (±0.52)	1.90 (±0.53)	1.88 (±0.52)	0.83
cMets	0.01 (±3.05)	0.00 (±3.15)	0.01 (±2.97)	0.99
SPISE	9.70 (±2.49)	9.80 (±2.73)	9.62 (±2.25)	0.59
Household income, n (%)				
<3 million KRW	16 (7.2)	7 (3.2)	9 (8.1)	0.89
3~5 million KRW	64 (28.8)	33 (14.9)	31 (14.0)	
≥5 million KRW	142 (64.0)	70 (31.5)	72 (32.4)	
Mother’s education level ^b^, n (%)				
Low	43 (19.6)	18 (8.2)	25 (11.4)	0.40
High	177 (80.5)	89 (40.5)	88 (40.0)	
Parental history of hypertension, n (%)				
No	168 (74.0)	87 (38.3)	81 (35.7)	0.17
Yes	59 (26.0)	24 (10.6)	35 (15.4)	
Frequency of overeating, n (%)				
None	115 (51.3)	51 (22.8)	64 (28.6)	0.19
1~2 times/week	89 (39.7)	46 (20.5)	43 (19.2)	
≥3 times/week	20 (8.9)	13 (5.8)	7 (3.1)	
Moderate physical activity, n (%)				
Never	42 (18.8)	14 (6.3)	28 (12.5)	0.01
1~2 times/week	101 (45.1)	46 (20.5)	55 (24.6)	
3~4 times/week	61 (27.2)	34 (15.2)	27 (12.0)	
≥5 times/week	20 (8.9)	15 (6.7)	5 (2.2)	

WC, waist circumference; BMI, body mass index; MAP, mean arterial pressure; SBP, systolic blood pressure; DBP, diastolic blood pressure; FBG, fasting blood glucose; TG, triglyceride; HDL-C, high-density lipoprotein-cholesterol; PsiMS, Pediatric simple metabolic syndrome score; cMetS, continuous metabolic syndrome score; SPISE, single-point insulin sensitivity estimator; MetS, metabolic syndrome. Numerical values represent the mean (± standard deviation), while categorical values are represented by n (%). Numbers may not sum to the total due to missing data. ^a^ Log transformation was performed because it was not normally distributed. ^b^ High school graduation of lower and university and graduate school graduation or higher.

**Table 3 metabolites-13-00133-t003:** Mean difference in the metabolic syndrome index in adolescence according to childhood overweight and sarcopenic obesity.

		Metabolic Syndrome Index		
Index		PsiMS	cMetS	SPISE
		Mean ± SD	Mean ± SD	Mean ± SD
By BMI ^a^	Normal (n = 200)	1.84 ± 0.50 *	−0.33 ± 2.86 **	10.05 ± 2.38 **
	Overweight (n = 27)	2.21 ± 0.59 *	2.49 ± 3.32 **	7.10 ± 1.69 **
By MFR ^b^	Normal (n = 131)	1.85 ± 0.46 *	−0.12 ± 2.55 *	10.08 ± 2.40 **
	Sarcopenic obesity (n = 23)	2.13 ± 0.56 *	1.84 ± 3.27 *	7.50 ± 2.01 **

PsiMS, pediatric simple metabolic syndrome score; cMetS, continuous metabolic syndrome score; SPISE, single-point insulin sensitivity estimator; SD, standard deviation; BMI, body mass index; MFR, muscle fat ratio. All indexes showed statistically significant mean differences between groups. ^a^ Subjects were classified as normal or overweight, with the 85th percentile as the cut-off point. ^b^ Sarcopenic obesity was defined as 1.25 for boys and 1.1 for girls. * *p* < 0.01. ** *p* < 0.001.

**Table 4 metabolites-13-00133-t004:** Multiple linear regression results for the effect of childhood overweight and sarcopenic obesity groups on the metabolic syndrome index in adolescence.

MetS Index	Overweight (Ref. Normal)				Sarcopenic Obesity (Ref. Normal)			
	Crude		Adjusted ^a^		Crude		Adjusted ^a^	
	*β* (95% CI)	*p*-Value	*β* (95% CI)	*p*-Value	*β* (95% CI)	*p*-Value	*β* (95% CI)	*p*-Value
PsiMS	0.37 (0.16, 0.58)	0.001	0.32 (0.11, 0.53)	0.003	0.28 (0.06, 0.49)	0.011	0.30 (0.08, 0.52)	0.009
cMetS	2.82 (1.64, 4.00)	<0.001	2.63 (1.45, 3.82)	<0.001	1.96 (0.77, 3.15)	0.002	1.96 (0.73, 3.20)	0.002
SPISE	−2.95 (−3.88, −2.02)	<0.001	−2.64 (−3.52, −1.75)	<0.001	−2.59 (−3.63, −1.54)	<0.001	−2.37 (−3.44, −1.31)	<0.001

MetS, metabolic syndrome; 95% CI, 95% confidence intervals; PsiMS, pediatric simple metabolic syndrome score; cMetS, continuous metabolic syndrome score; SPISE, single-point insulin sensitivity estimator; MetS, metabolic syndrome. ^a^ Adjusted for sex, monthly household income, the mother’s education level, and the parental history of hypertension, overeating, and moderate physical activity at the age of 13–15 years.

## Data Availability

The cohort data are not freely available due to privacy or ethical restrictions, but the Ewha Birth and Growth Study team welcomes collaborations with other researchers. For further information, contact Hyesook Park (the corresponding author).

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
