# Peer review of "The Effect of Childhood Obesity or Sarcopenic Obesity on Metabolic Syndrome Risk in Adolescence: The Ewha Birth and Growth Study"

_metabolites, 2023, doi:10.3390/metabo13010133_

Round 1

Reviewer 1 Report

I would like to thank the authors for their work. The paper is in general well-written, although it would benefit of an English spelling check. 

I would like to make some comments:

General comments:

The people-first language is not considered in this paper. The authors should read the EASO’s recommendations and avoid using language such “obese group”, “obese people”. Please change it to “Group with obesity”, “people living with obesity”. Other examples are welcome. 

Just a note: There are different letter’s sizes throughout the text. From line 47 to 48, and then from 64 to 65.

Line 56 – Appear obese… Not appropriate. 

Table 1 – Why the 13-15 group do not have all values?

Table 1 – How many boys and girls are involved for both groups? Are the % of girls the same for both groups?

Line 171 – Borderline statistical significance. Mathematically this is not a correct affirmation. Please read this paper and change these types of affirmations throughout the result’s section:

https://www.ncbi.nlm.nih.gov/pmc/articles/PMC6440716/

Line 174 – “Significant difference”. There is no need to use the word “significant”. If there is a difference, it is because it is statistically significant, otherwise no differences would be found. Please remove it.

Author Response

Dear Reviewer,

We appreciate you and the reviewers for your precious time in reviewing our paper and providing valuable comments. We have revised our manuscript incorporating these valuable suggestions and believe that our manuscript has been much improved.

Below we provide the point-by-point responses. All modifications in the manuscript have been highlighted in yellow.

Kind regards,

Reviewer 2 Report

This article by Park et al evaluate the impact of childhood obesity or sarcopenic obesity on the risk of adolescence metabolic syndrome. The study is well conducted and clearly reported.

Some minor concerns:

How was overeating defined in this study? Please elaborate.

List this as a limitation: “This study did not evaluate Insulin Sensitivity Index (ISI), Homeostasis Model Assessment for Insulin Resistance (HOMA-IR), or secretion (HOMA-β%) indicators that estimate insulin sensitivity and resistance” at the end of the paper.

Consider replacing “sex” by “gender”.

Page 7, lines 3-4: “The SIPSE level was also significantly lower”- please provide the statistically significant value. Same for lines 5-6.

Author Response

(The authors gave the same response as above.)
